# Anti-Colorectal Cancer Effects of a Novel Camptothecin Derivative PCC0208037 *In Vitro* and *In Vivo*

**DOI:** 10.3390/ph16010053

**Published:** 2022-12-30

**Authors:** Min Li, Linxu Wang, Yingjie Wei, Wenyan Wang, Zongliang Liu, Aixia Zuo, Wanhui Liu, Jingwei Tian, Hongbo Wang

**Affiliations:** 1School of Pharmacy, Key Laboratory of Molecular Pharmacology and Drug Evaluation (Yantai University), Ministry of Education, Collaborative Innovation Center of Advanced Drug Delivery System and Biotech Drugs in Universities of Shandong, Yantai University, Yantai 264005, China; 2Bohai Rim Advanced Research Institute for Drug Discovery, Yantai 264000, China; 3Luye Pharma Group, Yantai 264670, China

**Keywords:** PCC0208037, topoisomerase I, camptothecin, diarrhea, SN-38

## Abstract

Colorectal cancer is one of the most common malignancies, and the topoisomerase inhibitor irinotecan (CPT-11)-based chemotherapeutic regimen is currently the first-line treatment with impressive therapeutic efficacy. However, irinotecan has several clinically significant side effects, including diarrhea, which limit its clinical utility and efficacy in many patients. In an effort to discover better and improved pharmacotherapy against colorectal cancer, we synthesized a novel topoisomerase inhibitor, PCC0208037, examined its anti-tumor efficacy and related molecular mechanisms, and characterized its toxicity and pharmacokinetic profiles. PCC0208037 suppressed colorectal cancer cell (CRC) proliferation and increased cell cycle arrest, which may be related to its effects on up-regulating DNA damage response (DDR)-related molecules and apoptosis-related proteins. PCC0208037 demonstrated robust anti-tumor activity *in vivo* in a colorectal cancer cell xenograft model, which was comparable to or slightly better than CPT-11. In a preliminary toxicology study, PCC0208037 demonstrated much weaker tissue damage to colorectal tissue than CPT-11, and its impacts on food intake and body weight loss were more transient and recovered faster than CPT-11 in mice. This could be partially explained by the pharmacokinetic findings, which showed that PCC0208037 and its active metabolite, SN-38, were more accumulated in tumor tissue than in the intestine, as compared to CPT-11. Taken together, these results described a novel Topo I inhibitor with a comparative advantage over the standard treatment of colorectal cancer CPT-11 and could be a promising candidate compound for the treatment of colorectal cancer that warrants further investigation.

## 1. Introduction

Colorectal cancer is one of the most common malignancies as a long-term threat to human health and ranks third in terms of incidence and second in terms of mortality [1]. Chemotherapy, in combination with surgery, currently is the standard treatment strategy, which has been shown to significantly improve the 5-year survival rate [2,3]. According to the NCCN guidelines, the commonly used chemotherapy regimens most likely include a combination of two or three chemotherapeutic drugs, in which topoisomerase inhibitors, such as irinotecan, are a key component [4]. However, several clinically significant side effects related to topoisomerase inhibitors, including irinotecan, such as diarrhea, limit its therapeutic efficacy and clinical use [5,6]. Therefore, the development of novel agents, which could maintain irinotecan-like anticancer efficacy with less severe side effects, is a crucial drug discovery task that may substantially improve the therapeutic benefits to colorectal cancer patients.

Irinotecan (CPT-11) is a camptothecin derivative that was approved by the Food and Drug Administration in 1996 for the treatment of advanced colorectal cancer [7]. Irinotecan is one of the best-studied topoisomerase I inhibitors with significant anticancer efficacy via several possible mechanisms of action, including the formed ternary irinotecan-topoisomerase I-nicked DNA complex disabling the relegation of the nicked strand and prevention of topoisomerase release, which eventually leads to replication form arrest and cell death, and irinotecan acting as an efficient radiosensitizer. Once absorbed, carboxylesterases present in the body and will break the carbamate bond between the camptothecin moiety and the dipiperidino side chain of irinotecan to produce SN-38, the major active metabolite of irinotecan, which has significantly improved solubility and bioavailability. SN-38 will then repress cancer cell proliferation by inducing DNA damage by inhibiting the activity of topoisomerase I (Topo I) [8]. SN-38 will be transformed in the liver through glucuronidation by uridine diphosphate-glucuronosyltransferase (UGT) to form inactive SN-38G. After excreted via the biliary ducts into the gastrointestinal (GI) tract, SN-38G can be regenerated into SN-38 by the bacterial enzyme β-glucuronidase (GUS), which results in toxicity to the intestinal cells causing mucositis and diarrhea [9]. Given the high incidence of chemotherapy-induced diarrhea (as high as 82%), which substantially limits the clinical use of some effective anticancer drugs, including irinotecan, there have been efforts to develop a separate add-on pharmacotherapeutic approach to decrease the accumulation of SN-38 in the gastrointestinal system to reduce the occurrence of diarrhea with some success [5,10].

Given the well-validated anti-cancer target of Topo I and the clinical success of irinotecan, great efforts have been made to explore different strategies to design irinotecan (also known as CPT-11) analogs in the hope of reducing its side effects or/and improving its therapeutic efficacy, with several promising compounds reported, including F10 [11], BQC-Glucuronide [12], and ZBH-1205 [13].

In this study, we reported our own effort of developing a novel camptothecin derivative, designated as PCC0208037 (Figure 1), which demonstrated superior anti-colorectal cancer activity both *in vivo* and *in vitro* with reduced accumulation of SN-38. These results support PCC0208037 as a novel lead compound with strong anti-tumor efficacy and reduced diarrhea side effects as compared to irinotecan.

## 2. Results

### 2.1. PCC0208037 Suppressed the Proliferation of Colorectal Cancer Cells

To examine the effects of PCC0208037 on colorectal cancer cells, a growth inhibition assay was performed with PCC0208037 in four lines of colorectal cancer cells (LS180, HCT116, HT-29, and CT-26). The MTT results showed that PCC0208037 demonstrated significant and dose-dependent cytotoxic activity against LS180, HCT116, HT-29, and CT-26 cells. The IC_50_ values of PCC0208037 against LS180, HCT116, HT-29, and CT-26 were 0.22, 2.90, 2.33, and 0.66 µmol/L, respectively (Table 1). PCC0208037 showed weaker cytotoxic activity than SN-38 and stronger cytotoxic activity than CPT-11.

In the colony formation assay, PCC0208037 significantly inhibited colony formation (Figure 1). In the LS180 cells, the colony formation rates at doses of 0.04 and 0.08 µmol/L of PCC0208037 were 21.56% and 41.96%, respectively. In the HCT116 cells, the colony formation rates at doses of 0.04 and 0.08 µmol/L of PCC0208037 were 33.15% and 49.72%, respectively. These results suggested that PCC0208037 has good anti-proliferative effects *in vitro*. Consistent with these findings, PCC0208037 decreased the cell growth ability in a dose-dependent manner in the cell growth curve assay (Figure 2).

### 2.2. Molecular Docking of PCC0208037

To understand why PCC0208037 exhibited a weaker inhibitory effect than SN-38 on Topo I, optimal molecular docking (CDOCKER, Discovery Studio 2018) was performed. Ten replicas for PCC0208037 and SN-38 were produced as a spherical scope with a diameter of 18 Å and centered on the Topotecan region (Table 2). From the complex conformation between PCC0208037 and SN-38 (Figure 3), it was found that the two molecules were surrounded by the residues ASN722, ASN352, ASP533, ILE535, and ARG364. The hydroxy group on the benzene ring of PCC0208037 formed a hydrogen bond with Glu356. In addition, the O atom on the lactone ring of PCC0208037 formed another hydrogen bond with THR718. However, PCC0208037 did not show key polar interactions with ASP533, which might help explain why SN-38 showed a stronger inhibitory effect than PCC0208037 in the Topo I enzyme assay.

### 2.3. PCC0208037 Inhibited Topo I Enzyme Activity

To determine whether PCC0208037 inhibited Topo I activity, a DNA relaxation assay was conducted. In this assay, supercoiled plasmid DNA was relaxed and nicked by recombinant Topo I, while no nicked DNA would be found when the Topo I enzyme was inhibited. CPT-11, as the prodrug of SN-38, showed similar results as a vehicle. In contrast, the bioactive metabolite of the CPT-11 SN-38 treatment led to clear supercoiled DNA, suggesting that Topo I activity was inhibited by SN-38. Similarly, PCC0208037 dose-dependently increased the presence of supercoiled DNA, suggesting that PCC0208037 could inhibit Topo I activity (Figure 4).

### 2.4. PCC0208037 Induced G_2_/M Phase Cell Cycle Arrest and Cell Apoptosis

To explore the effect of PCC0208037 on cell cycle progression, LS180 cells were treated with PCC0208037 and then examined by FCM flow cytometry. The treatment with PCC0208037 significantly increased the ratio of the cells in the G_2_/M phase over 24, 48, and 72 h treatments. Similarly, double staining with FITC annexin V showed that 24, 48, and 72 h treatments with PCC0208037 significantly increased the percentage of apoptotic cells (Figure 5).

### 2.5. Effects of PCC0208037 on Signaling Molecules Related to DNA Damage and Apoptosis in CRC Cells

The effects of PCC0208037 on the key signaling of the molecules associated with DNA damage were studied in LS180 and HCT116 cells (Figure 6). Compared to the control group, 8 h of PCC0208037 treatment significantly up-regulated the protein levels of γ-H2AX, p-ATM, p-Chk1, p-Chk2, p-P53, and P53.

The effects of PCC0208037 on key signaling molecules associated with apoptosis were investigated in LS180 and HCT116 cells (Figure 7). Compared to the control group, 24 h of PCC0208037 treatment significantly up-regulated the protein levels of BAX and PUMA and down-regulated the protein level of Bcl-2.

### 2.6. PCC0208037 Demonstrated Good Anti-Tumor Activity In Vivo

Based on the *in vitro* inhibitory effects against colorectal cancer cells, we further assessed the anti-tumor efficacy of PCC0208037 in a subcutaneous LS180 xenograft model. Compared to the control, PCC0208037 (5, 10, and 20 mg/kg) treatment led to a significant and dose-dependent reduction in the tumor weight. The tumor inhibition rates were 29.9%, 37.2%, and 50.8%, respectively, for the three doses of PCC0208037. Importantly, PCC0208037 achieved this anti-tumor activity without impacting the normal feeding and body weight gain in mice. In contrast, animals that received the CPT-11 treatment exhibited a significant reduction in body weight (Figure 8).

### 2.7. Preliminary Toxicological Evaluation of PCC0208037 in Mice

No animal died during the seven-day observation period. Starting from the 2nd day, the body weight and food consumption began to decrease in mice that received 50 mg/kg of CPT-11 or the 50 mg/kg PCC0208037 treatment. Interestingly, the body weight and food intake decrease began to recover on the 6th day onwards in mice that received the 50 mg/kg PCC0208037 treatment, while no apparent trend of recovery was observed in mice that received 50 mg/kg of CPT-11. The diarrhea score was also higher in the CPT-11-treated mice than in the PCC0208037-treated mice, particularly during the later stage of the study period. The divergence between the two groups reached statistical significance on the 7th day in body weight change, the amount of food intake, and the diarrhea score (*p* < 0.05; Figure 9), suggesting a relatively quick recovery in the PCC0208037-treated mice.

A colorectal tissue histomorphological examination revealed visually noticeable differences between the PCC0208037-treated and CPT-11-treated mice. In particular, the microstructure of the colorectal tissue in the control and PCC0208037-treated mice demonstrated better integrity compared to the CPT-11-treated mice. CPT-11 caused extensive colorectal tissue damage, as evidenced by epithelial injury and severe inflammatory infiltration. A mild reduction of goblet cells and a mild infiltration of lymphocytes were observed in the crypt of the PCC0208037-treated mice (Figure 9).

### 2.8. Pharmacokinetics and Tissue Distribution Study

The pharmacokinetics study showed that the plasma exposure of PCC0208037 was far higher than that of CPT at equimolar doses greater than 100-fold (Table 3). PCC0208037 also demonstrated better absorption and faster onset of action than CPT-11. Consistent with this difference, the concentration of the major metabolite of both compounds, SN-38, was at least 2-fold higher in the PCC0208037-treated than in the CPT-11-treated mice (Table 3). However, The T_1/2_ of PCC0208037 was relatively short, which may be related to the rate at which the metabolite was released by the prodrug metabolism.

The tissue distribution results showed that the concentration of PCC0208037 in the colorectal tissue was significantly lower than CPT-11 at 0.25 h and 1 h after the drug’s administration (Figure 10). Interestingly, although the concentration of SN-38 in the colorectal tissue was not different between the PCC0208037-treated and CPT-11-treated mice, the concentration of SN-38 in the tumor tissue was much higher (~6-fold difference) in the former mice, at 0.25 h and 1 h after the drug’s administration (Figure 10).

## 3. Discussion

Colorectal cancer is one of the most life-threatening malignancies worldwide [14]. CPT-11, a camptothecin derivative, is a standard first-line drug for treating advanced CRC in the clinic with good efficacy [15]. However, the clinical use of CPT-11 is substantially limited by its accompanying side effects, such as diarrhea [16]. Efforts have been made to mitigate and control late-onset diarrhea in patients who receive CPT-11 treatment for colorectal cancer, with mixed results [17,18]. Here, we report for the first time that PCC0208037, a novel inhibitor of Topo I, demonstrated significant anti-tumor efficacy, which was comparable or slightly superior to CPT-11 both in *in vitro* and *in vivo* tests but with substantially reduced toxicity in the gastrointestinal system.

We synthesized PCC0208037 in-house and, as a first step, evaluated its anticancer efficacy against three lines of colorectal cancer cells, which showed that PCC0208037 was more effective in inhibiting cell proliferation than CPT-11 *in vitro*. It is well known that Topo I plays a vital role during the transcription and replication processes. Camptothecin derivatives can bind to DNA-Topo I to form a stable ternary complex, which inhibits the activity of Topo I. When this ternary complex encounters the replication fork, it leads to double-stranded DNA damage [19]. One consequence of DNA damage is the induction of cell cycle arrest. PCC0208037 is a CPT derivative and an inhibitor of Topo I, as evidenced by the molecular docking and Topo I activity assay results. Not surprisingly, the cell cycle arrest in the G_2_/M phase was clearly demonstrated in PCC0208037-treated colorectal cancer cells. Taken together, these results suggest that PCC0208037 is a Topo I inhibitor and could inhibit colorectal cancer via the induction of cell cycle arrest.

After this initial exploration, we continued to understand how PCC0208037 works mechanistically to produce anticancer activity *in vitro*. Camptothecin derivatives are known to trigger the response of several molecular pathways, including the cell cycle checkpoint pathway, which involves multiple checkpoint kinases, such as ATM, ATR, and their relays, CHK1 and CHK2, the activation of which leads to delayed cell cycle progression [20]. We examined whether PCC0208037 could affect the cell cycle checkpoint pathway, and a Western blotting assay revealed that the expression levels of multiple key molecules, such as DDR, γ-H2AX, ATR, p-ATR, ATM, p-ATM, Chk1, p-Chk1, Chk2, and p-Chk2, were significantly up-regulated by the PCC0208037 treatment. These results suggest that PCC0208037-induced cell cycle arrest could be mediated via the ATR/ATM-Chk1/Chk2 pathway.

Unrepaired DNA strand breaks can activate an apoptosis pathway, which leads to cell death. Camptothecin derivatives are known to induce apoptosis in a p53-dependent manner. Because p53 upregulates genes, such as PUMA and BAX, and downregulates genes, such as Bcl-2, to promote cell death through apoptosis in the DNA damage response (DDR) [21], we next examined whether PCC0208037 affects these key apoptosis-related molecules. Indeed, the PCC0208037 treatment significantly upregulated the expression levels of the proteins involved in the DDR, such as p53, p-p53, PUMA, and BAX, and down-regulated the expression level of Bcl-2. These results suggest that, like other camptothecin derivatives, PCC0208037 induced apoptosis in a p53-dependent manner.

After the mechanistic exploration, we next examined the *in vivo* anti-tumor efficacy of PCC0208037 using a classical xenograft cancer model. Consistent with the *in vitro* findings, PCC0208037 was able to significantly suppress the growth of tumors derived from LS180 cells *in vivo* in a dose-dependent manner. In terms of anti-tumor efficacy, PCC0208037 was at least comparable, if not superior, to CPT-11. Given that CPT- 11 has well-established side effects on the gastrointestinal system, we also performed a preliminary toxicological assessment of PCC0208037. The results found that at an ultra-high dose, PCC0208037 reduced food intake, which led to body weight loss on a level similar to the equivalent dose of CPT-11, suggesting that PCC0208037 also has side effects on the gastrointestinal system. However, it seems that these side effects were shorter-lived and recovered quickly as compared to CPT-11. A histomorphological examination of the colorectum tissue revealed much milder tissue damage and local inflammation in the PCC0208037-treated mice than in the CPT-11-treated mice. These results suggest that PCC0208037 has comparable anti-tumor efficacy to CPT-11 but a substantially lower colonic toxicity than CPT-11 in mice.

Finally, we evaluated the pharmacokinetics and tissue distribution profiles of PCC0208037 in mice to better understand its drug-like property. The pharmacokinetics results show that the exposure of PCC0208037 was far higher (> 100 times) than that of CPT-11, and the exposure of the major metabolite of both compounds, SN-38, was also at least twice as high for PCC0208037 than for CPT-11. The tissue distribution results showed that PCC0208037 could be easily metabolized into SN-38 *in vivo*, and the concentration level was much higher in the tumor tissue than in the colorectum, as compared to the CPT-11 treatment, which could explain the lower enterotoxicity of PCC0208037 than CPT-11.

## 4. Materials and Methods

### 4.1. Drugs and Chemicals

The 1 H NMR (400 MHz) and 13 NMR (100 MHz) spectra were recorded on a Bruker AV-400 spectrometer. Chemical shifts were reported in parts per million (ppm) and tetramethylsilane as an internal standard. LC-MS/MS analysis was performed on an LCMS-8040 tandem quadrupole mass spectrometer (Shimadzu, Kyoto, Japan). The crude product was purified using silica gel column flash chromatography. The purity of the compounds was ≥95% by high-pressure liquid chromatography (HPLC).

### 4.2. Synthesis of PCC0208037

A solution of SM1 (200 mg, 0.51 mmol, and 1.0 eq) in toluene (5 mL) was added into the SM2 (335.7 mg, 1.02 mmol, and 2.0 eq) and triphosgene (302 mg, 1.02 mmol, and 2.0 eq). The reaction mixture was stirred at 120 °C for 3 h, and 60% of the desired product was found on LC-MS. Then, 26 batches were made with an identical process and processed together. The solvent was removed to obtain a 14.9 g yellow crude product, which was submitted to purify by gel column chromatography (MeOH: DCM = 2:100 to 3:100) to obtain 3.2 g of a yellow solid product.

A solution of PCC0208037-1 (3.2 g, 4.3 mmol, and 1.0 eq) in MeOH (50 mL) was added with a 1N NaOH solution (12.9 mL, 12.9 mmol, and 3.0 eq) at 0 °C. The resulting solution was stirred at 0 °C for 2 h. We adjusted the mixture to a PH = 7 with HCl and dried it with Na_2_SO_4_. It was concentrated and purified by gel column chromatography (MeOH: DCM = 3:100 to 5:100) to obtain the final 1.1 g yellow solid product of PCC0208037.

LC-MS m/z: 664.4.2 (M-H) +; RT = 1.165 min (2.5 min).

1H NMR (400 MHz, DMSO-d6) δ 11.63 (s, 0H), 11.16 (s, 1H), 10.32 (s, 1H), 8.03 (dd, J = 11.6, 7.3 Hz, 2H), 7.40 (dd, J = 6.3, 2.7 Hz, 2H), 7.01 (s, 1H), 5.61 (d, J = 2.6 Hz, 1H), 5.51 (s, 2H), 5.37 (d, J = 22.7 Hz, 1H), 5.30 (s, 2H), 5.09–4.87 (m, 1H), 3.99 (s, 1H), 3.88 (s, 1H), 3.63 (s, 1H), 3.08 (q, J = 7.5 Hz, 2H), 2.16 (*p*, J = 7.0 Hz, 2H), 1.29 (t, J = 7.5 Hz, 6H), 0.95 (d, J = 25.7 Hz, 3H).

HPLC: purity @254 nm: 96.9%; 214 nm: 97.5%.

### 4.3. Materials

Antibodies against ATM (2873), p-ATM (13050), ATR (13934), p-ATR (2853), Chk1 (2360), p-Chk1 (2348), Chk2 (2662), p-Chk2 (2348), P53 (18032), p-P53 (9284), Bcl2 (15071), β-actin (3700), and GAPDH (5174) were obtained from Cell Signaling Technology (Danvers, MA, USA). Gamma H2AX (ab26350) and Bax (ab32503) were obtained from Abcam (Cambridge, UK). Puma (p4743) was obtained from Sigma-Aldrich (St. Louis, MO, USA). Methylthiazolyldiphenyl-tetrazolium bromide (MTT) (ST316), a penicillin streptomycin solution (C0222), an enhanced BCA protein assay kit (P0009), an ultra-hypersensitive ECL chemiluminescence kit (P0018AS), and a crystal violet staining solution (C0121) were obtained from Beyotime Biotechnology (Shanghai, China). An RPMI 1640 medium (11875093), McCoy’s 5A medium (16600082), and fetal bovine serum (10099141) were obtained from Thermo-Fisher (Waltham, MA, USA). A Topoisomerase I Drug Screening Kit (TG1018) was obtained from TopoGEN, Inc (Buena Vista, CO, USA). A FITC Annexin V Apoptosis Detection Kit I (556547) was obtained from BD Biosciences (Franklin Lakes, NJ, USA). A Cell Cycle Detection Kit (KGA512) was obtained from KeyGEN Biotech (Nanjing, China). SN-38 (BCP01386) and CPT-11 (BCP02860) were purchased from ShangHai Biochempartner Co., Ltd. (Shanghai, China). PCC0208037 was dissolved in DMSO and stored at −20 °C for further use in *in vitro* experiments. For *in vivo* experiments, PCC0208037 was dissolved in a 1:1 solution of ethanol and cremophor and then diluted in saline immediately before use.

### 4.4. Cell Lines and Animals

The colorectal cancer cell lines LS180, HCT116, CT-26, and HT-29 were purchased from the Type Culture Collection of the Chinese Academy of Sciences (Shanghai, China). LS180 and CT-26 cells were cultured in an RPMI 1640 medium, and HCT116 and HT-29 cells were cultured in McCoy’s 5A medium, supplemented with 10% fetal bovine serum (FBS) and 1% penicillin-streptomycin. The colorectal cancer cell lines were all cultured in a humidified incubator at 37 °C with 5% CO_2_.

Athymic nude mice (6–8 weeks old, BALB/c, male) were purchased from Vital River Laboratory Animal Technology (Beijing, China). Mice (6–8 weeks old, BALB/c, male) were purchased from the Jinan Pengyue Laboratory Animal Breeding Co., Ltd. Mice were housed in a sterile environment, and they had free access to drinking water and standard rodent chow. Mice were acclimated to the new housing environment for a week before any experiment. The housing room was maintained under a 12 h light/dark cycle at 21 °C room temperature and 55% humidity. The Experimental Animal Ethics Committee of the Yantai University of China approved the study. Animals were maintained and experiments were conducted in accordance with the 2011 Guide for the Care and Use of Laboratory Animals (the Institute of Laboratory Animal Resources on Life Sciences, National Research Council, National Academy of Sciences, Washington, DC, USA).

### 4.5. Cell Viability Assay

Cell viability was assessed using the MTT assay, as previously reported [22]. Briefly, colorectal cancer cells (LS180, HCT116, CT-26, and HT-29 cells) were plated onto 96-well plates at 3000 cells per well and incubated overnight. Cancer cells were treated with different concentrations of PCC0208037, SN-38, or CPT-11 (20, 6, 2, 0.3, 0.1, 0.03, 0.01, 0.003, and 0.001 µmol/L). Control cells were treated with DMSO at a final concentration of 0.1%. Cells were cultured with the compounds or medium for 72 h. The cells were then treated with MTT. The medium was then decanted, and DMSO was added and shaken until the color reaction was complete. The negatives were read using a Molecular Devices SpectraMax M5 (San Jose, CA, USA) at OD_570_ nm. The half maximal inhibitory concentration (IC_50_) was calculated.

### 4.6. Cell Proliferation Assay

Cell proliferation was examined by a cell growth curve assay [23] and a clone formation assay [24]. Colorectal cancer cells (LS180 and HCT116 cells) were plated onto six-well plates at a density of 8000 cells per well. Twenty-four h later, cells were treated with PCC0208037, SN-38, or CPT-11 at various concentrations (PCC0208037: 0.001, 0.003 µmol/L; SN-38: 0.003 µmol/L; CPT-11: 0.3 µmol/L). The number of cells was measured 24 h after the drug treatments. Growth curves were generated from cell counts over 6 consecutive days. For the clone formation assay, colorectal cancer cells (LS180 and HCT116 cells) were plated onto six-well plates at a density of 800 cells per well. Twenty-four h later, cells were treated with PCC0208037, SN-38, or CPT-11 at various concentrations (PCC0208037: 0.04, 0.08 µmol/L; SN-38: 0.02 µmol/L; CPT-11: 0.3 µmol/L). Cells were then incubated with designated concentrations of PCC0208037. Fourteen days later, cells were stained with a crystal violet solution, and clones were counted.

### 4.7. Molecular Docking Studies

The X-ray structure of the human Topo I-DNA covalent complex was obtained from the Protein Data Bank (PDB code:1K4T) [25] and reserved for docking-based studies. All water molecules were removed, hydrogen atoms were added to the complex, and hydrogen bonds were maximized by optimizing key residues. The 3D structures of PCC0208037 and SN-38 were drawn using ChemBioDraw followed by local minimization using the MMFF force field. This structure was used for subsequent docking simulations. First, the geometry of the protein was optimized using a fast DREIDING force field. Target proteins were detected using the Discovery Studio 2018 Clean Geometry toolkit. Ten replicas of each of the compounds, PCC0208037, and SN-38, with a diameter of 9 Å, were then produced and centered on the Human Topo I-DNA covalent complex adducts. The final 10 best conformations were reserved for further analysis.

### 4.8. Topoisomerase I Activity Assay

A Topoisomerase I Drug Screening Kit was used to detect the inhibitory activity of PCC0208037 against Topo I in the cell-free system [26]. 1 IU Topo I and 250 ng of supercoiled plasmid DNA were pre-incubated with DMSO, PCC0208037 (100 nmol/L and 10 nmol/L), SN-38(100 nmol/L), or CPT (100 nmol/L) in a reaction buffer with a final volume of 20 μL and incubated for 30 min at 37 °C; then, the reaction was stopped by the addition of 2 ul of 10% SDS. Then, proteinase K was added to 50 ug/mL, incubated for 15 min at 37 °C, and 0.1 vol. of a loading buffer (0.25% bromophenol blue, 50% glycerol) was added. The supercoiled, relaxed, or nicked DNA was separated by 1% agarose gel in a 1 TAE (Tris-Acetate-EDTA) buffer. Ethidium bromide-stained agarose gel was photographed using Gel Doc XR.

### 4.9. Flow Cytometry Assay

Cell cycle analysis and an Apoptosis Assessment were performed using flow cytometry [27]. LS180 cells were seeded in 6-well plates at a density of 1 × 10^5^ cells per well for 12 h, and then different concentrations of PCC0208037 were added for 24, 48, and 72 h, respectively. Cells were stained with PI- or PI/FITC-labeled annexin, and the effects of PCC0208037 on cell cycle progression and apoptosis were analyzed by flow cytometry.

### 4.10. Western Blot Assay

Effects of PCC0208037 on the protein expression levels were examined by Western blotting, as described previously [28]. Briefly, 2 × 10^6^ LS180 and HCT116 cells were seeded in 10-cm dishes to adhere overnight. Then, the cells were treated with DMSO (final concentration = 0.1%), PCC0208037 (0.01 μmol/L and 0.03 μmol/L), SN-38 (0.01 μmol/L and 0.03 μmol/L), or CPT-11 (3 μmol/L and 10 μmol/L) for 8 or 24 h. At the desired time points, cells were washed twice with a cold PBS. Then, cells were lysed in the RIPA buffer for 30 min on ice. The RIPA buffer contains protease inhibitors and phosphatase inhibitors. After sonication, the samples were centrifuged at 12,000 rpm, and then the supernatants were harvested and assayed for the protein concentration. Equal amounts of protein (30 μg) were separated with a polyacrylamide gel and transferred to a PVDF membrane. The membranes were blocked for 2 h (in 5% skim milk) and then incubated with primary antibodies overnight at 4 °C: ATM p-ATM, ATR, p-ATR, Chk1, p-Chk1, Chk2, p-Chk2, P53, p-P53, Bcl2, gamma H2AX, Bax, Puma, β-actin, and GAPDH. Then, the membranes were incubated with secondary antibodies for 2 h at room temperature. All these antibodies were diluted to a 1:1000 concentration in use. The immunoblots were visualized by a BeyoECL Plus enhanced chemiluminescence system.

### 4.11. Colorectal Cancer Xenograft Model

To establish xenograft tumor models, RPMI 1640 and Matrigel were mixed at a 1:1 ratio. Colorectal cancer cells (LS180: 5 × 106 cells) were added to 0.1 mL of the mixture, and male nude mice were subcutaneously injected with colorectal cancer cells on the right side [29]. When the size of the tumors reached a volume of roughly 130 × 150 mm^3^, as measured by calipers (volume = 1/2 (Length × Width^2^)), the mice were randomly assigned to one of the following five groups (*n* = 8): (a) vehicle (polyethylene castor oil and ethanol, 1:1; dilute with normal saline, 1:3, before administration.); (b) 5 mg/kg PCC0208037; (c) 10 mg/kg PCC0208037; (d) 20 mg/kg PCC0208037, and (e) 10 mg/kg CPT-11. PCC0208037 and CPT-11 were intravenously administered once every 3 days. Tumor volume and animal weight were measured every 3 days and continued until the end of the experiments. All mice were euthanized by cervical dislocation at the end of the experiment, and the xenograft tumors were removed and weighed.

### 4.12. PCC0208037- and CPT-11-Induced Toxicity in Mice

Mice were randomly assigned to one of the following three groups (*n* = 12/group): (a) vehicle (polyethylene castor oil and ethanol, 1:1; dilute with normal saline, 1:3, before administration.); (b) 50 mg/kg PCC0208037; (c) 50 mg/kg CPT-11; PCC0208037 was intraperitoneally administered daily (i.p.; 50 mg/kg/day) to mice for four consecutive days, based on previously reported method [30]. Mice were monitored daily for 7 days starting from the first day of drug administration. The degree of diarrhea, the body weight of the mice, and the amount of food intake were recorded. The degree of diarrhea in mice was scored using the following system [31]: 0, normal (normal or stool absent); (1) minor diarrhea (wet, soft stool); (2) moderate diarrhea (wet, unformed stool with moderate perianal staining of the coat); and (3) severe diarrhea (watery stool with severe perianal coat staining). After 7 days, the mice were euthanized, and the colons were removed and fixed in 4% paraformaldehyde and embedded in paraffin for histological analysis (H&E staining and morphological examination).

### 4.13. Pharmacokinetic Study and Tissue Distribution Study

The pharmacokinetics of PCC0208037 and CPT-11 were studied by a single injection protocol in BALB/c mice, with the drugs administered through the tail vein [14]. Mice were randomly divided into three groups (*n* = 3/group): control, 10.65 mg/kg of PCC0208037, and 10 mg/kg of CPT-11, in which PCC0208037 and CPT-11 were at equal molar dosages. Blood samples were collected at 0.083 h, 0.25 h, 0.5 h, 1 h, 2 h, 4 h, 8 h, and 12 h after administration. Plasma was prepared after centrifugation at 6000× *g* for 5 min and then stored at −20 °C for further analysis. The concentrations of PCC0208037, CPT-11, and their metabolite, SN-38, in mice plasma were extracted using liquid–liquid extraction and analyzed using the LC-MS/MS (Thermo Electron Corporation, San Jose, CA, USA) method.

BALB/c xenografted nude mice were given a single injection of PCC0208037 or CPT-11 through the tail vein. Then, 0.25 h and 1 h after the injection, the colorectal and tumor tissue were removed, rinsed with saline, and blotted dry. Colorectal and tumor tissues were then homogenized with saline (1:4, *w*/*v*) and the concentrations of A, CPT-11, and their metabolite, SN-38, were extracted using liquid–liquid extraction and analyzed using the LC-MS/MS (Thermo Electron Corporation, San Jose, CA, USA) method.

## 5. Conclusions

This study synthesized and characterized a novel camptothecin derivative, PCC0208037. *In vitro* studies confirmed the anti-tumor efficacy of PCC0208037 and explored the potential molecular mechanism of its anti-tumor efficacy: cell cycle arrest and DNA damage. *In vivo* studies further confirmed its comparable anti-tumor efficacy, as compared to CPT-11, in a widely-used xenograft cancer model and its lower enterotoxicity at an ultra-high dose. A pharmacokinetics evaluation revealed high plasma exposure and tumor tissue concentration of the PCC0208037 active metabolite, SN-38, supporting a more favorable drug-like property than CPT-11. Combined, these results suggest that PCC0208037 demonstrates a comparative advantage to the first-line colorectal cancer therapy, CPT-11, against colorectal cancer in preclinical studies and warrant further evaluation as a potential candidate drug to treat colorectal cancer.

## Data Availability

Data is contained within the article.

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
