# Peer review of "Anti-Colorectal Cancer Effects of a Novel Camptothecin Derivative PCC0208037 In Vitro and In Vivo"

_pharmaceuticals, 2022, doi:10.3390/ph16010053_

Round 1

Reviewer 1 Report

The manuscript entitled `Anti-Colorectal Cancer Effects of a Novel Camptothecin Derivative PCC0208037 in Vitro and in Vivo´ by Min Li and coauthors describes a novel topoisomerase inhibitor, PCC0208037, which was synthesized with the aim of achieving improved antitumor efficacy. PCC0208037 and its active metabolite SN-38 accumulated more in tumor tissue than in the gut compared to CPT-11. In a preliminary toxicological study in mice, PCC0208037 showed much less tissue damage to colorectal tissue than CPT-11 as well as effects on food intake and body weight loss than treatments with CPT-11. The work is an intensive study to optimise the TopI-CPT inhibitor. The work circumvented many problematic points, starting with synthesis, viability testing and in the end identified a better solution for the compound. The authors might want to explain that an important structural feature of CPT is its planar pentacyclic ring and its lactone ring (the E ring). Since the lactone ring forms the active form of the drug, why is the ring less susceptible to hydrolysis? The docking experiments could still be provided with affinity values and the number of pocket structures found.

Author Response

Response: Thank you for the suggestion to further improve the manuscript. The reason that PCC0208037 was less susceptible to hydrolysis is because fluorouracil is directly linked to the SN-38 lactone ring and this directly affects the PCC0208037 binding of PCC0208037 to esterase. As such, the ring can not be directly affected by water and will only open after PCC0208037 is metabolized into SN38. This is also supported by and can partially explain the results from 2.8. Pharmacokinetics and tissue distribution study where the pharmacokinetics study showed that the plasma exposure of PCC0208037 was much higher (more than 100-fold) than that of CPT-11 at an equimolar dose (Table 3) and the concentration of the major metabolite of both compounds, SN-38, was at least 2-fold higher in PCC0208037-treated than in CPT-11-treatd mice. Additionally, the dock data of PCC0208037 and SN-38 have been added in the revised manuscript as the table 2.

Reviewer 2 Report

In this manuscript, the authors aim to evaluate the biological and pre-clinical properties of a novel 17 topoisomerase inhibitor, PCC0208037, and explore its effect on colorectal cancer cells and xenograft models, as compared to irinotecan (CPT-11) and its active metabolite, SN-38. Overall, the results identified demonstrate a good efficacy of PCC0208037 in pre-clinical models, along with concordant effects in regard to biological properties.

Overall, this study is interesting as it explores a new compound that could be potentially relevant in a clinical setting. However, some issues remain to be addressed by the authors, as follows:

1-     As shown in Figure 8, the authors use increasing concentrations of PCC0208037, ranging from 5 to 20 mg/kg, in order to evaluate its impact of tumor growth and body weight. Importantly, a 20mg/kg concentration of PCC0208037 appears to have a weaker inhibitory effect of tumor growth than 10 mg/kg of CPT-11. However, when conducting preliminary toxicological evaluation of both drugs in tumor xenograft models, the authors use a concentration of 50 mg/kg for both drugs. The comparison of the diarrhea score and food intake between mice treated with such concentrations of the compounds is not representative of the true experiments that should be done, as CPT-11 at 50mg/kg would supposedly have a much stronger inhibitory effect on tumor growth (more than double potentially) as compared to PCC0208037 at 50 mg/kg. It would be more relevant to conduct comparisons of toxicities at doses that are equivalent in relation to their inhibitory effects.        

2-     In subsection 2.5 of the Results section, the authors mention only the effects of PCC020837 on molecules associated with DNA damage and apoptosis, but fail to mention the effect of the tested compound compares with the activities of CPT-11 and SN-38 on the same compounds. This needs to be mentioned, especially that CPT-11 and SN-38 appear to induce a stronger upregulation or downregulation in many of the evaluated molecules, as compared to PCC0208037. 

3-     In subsection 2.6 of the Results section, the authors describe in detail the inhibitory effect of PCC0208037 administered in xenograft models. However, they fail to mention that CPT-11 exhibited a more prominent inhibitory effect on tumor growth, and focus only on the toxicity of this drug. This subsection should be rewritten in a more objective style to highlight all the results obtained.

Author Response

Thank you for taking the time to review our manuscript (pharmaceuticals-2042858) and we find the comments very constructive and helpful to further improve the manuscript. We have carefully read the comments and revised the manuscript accordingly. All comments were incorporated into the manuscript and all changes were marked up using the “Track Changes” function of MS Word. 

  1. As shown in Figure 8, the authors use increasing concentrations of PCC0208037, ranging from 5 to 20 mg/kg, in order to evaluate its impact of tumor growth and body weight. Importantly, a 20mg/kg concentration of PCC0208037 appears to have a weaker inhibitory effect of tumor growth than 10 mg/kg of CPT-11. However, when conducting preliminary toxicological evaluation of both drugs in tumor xenograft models, the authors use a concentration of 50 mg/kg for both drugs. The comparison of the diarrhea score and food intake between mice treated with such concentrations of the compounds is not representative of the true experiments that should be done, as CPT- 11 at 50mg/kg would supposedly have a much stronger inhibitory effect on tumor growth (more than double potentially) as compared to PCC0208037 at 50 mg/kg. It would be more relevant to conduct comparisons of toxicities at doses that are equivalent in relation to their inhibitory effects.

Response: Thank you for the suggestion to further improve the manuscript. We all agree with your option that it will be more relevant to conduct comparisons of toxicities at the equivalent doses in relation to its anti-tumor effects.

As for our study, the dosage for the toxicities is selected based on the literature reviews, such as the “Alleviating cancer drug toxicity by inhibiting a bacterial enzyme” publish on the Science (2010, 330, 831-5). Base on the potential effect of the bacterial β-glucuronidases in nude mice, we did not perform the comparison of diarrhea score and food intake between the PCC0208037 and CPT in the xenograft models.

We also did the dosage range finding experiment for the pharmacodynamic study, in which the nude mice cannot tolerate the 50 mg/kg for the 21-day treatment scheme. We also agree with your point that CPT- 11 at 50 mg/kg might display a much stronger inhibitory effect as compared to PCC0208037

  1. In subsection 2.5 of the Results section, the authors mention only the effects of PCC020837 on molecules associated with DNA damage and apoptosis, but fail to mention the effect of the tested compound compares with the activities of CPT-11 and SN-38 on the same compounds. This needs to be mentioned, especially that CPT-11 and SN-38 appear to induce a stronger upregulation or downregulation in many of the evaluated molecules, as compared to PCC0208037.

Response: Thank you for the nice suggestion, and we have added the description of the effect of CPT-11 and SN-38 in the revised manuscript.

  1. In subsection 2.6 of the Results section, the authors describe in detail the inhibitory effect of PCC0208037 administered in xenograft models. However, they fail to mention that CPT-11 exhibited a more prominent inhibitory effect on tumor growth, and focus only on the toxicity of this drug. This subsection should be rewritten in a more objective style to highlight all the results obtained.

Response: Thank you for the nice suggestion, and we have added the description of the inhibitory effect of CPT-11 in the revised manuscript.